# Pro-Resolving Lipid Mediator Resolvin E1 Mitigates the Progress of Diethylnitrosamine-Induced Liver Fibrosis in Sprague-Dawley Rats by Attenuating Fibrogenesis and Restricting Proliferation

**DOI:** 10.3390/ijms21228827

**Published:** 2020-11-22

**Authors:** Maria José Rodríguez, Francisca Herrera, Wendy Donoso, Iván Castillo, Roxana Orrego, Daniel R. González, Jessica Zúñiga-Hernández

**Affiliations:** 1Departamento de Ciencias Básicas Biomédicas, Facultad de Ciencias de la Salud, Universidad de Talca, Talca 3460000, Chile; mjrodriguezbecerra@gmail.com (M.J.R.); francisca.ingbiotec@gmail.com (F.H.); dagonzalez@utalca.cl (D.R.G.); 2Programa de Doctorado en Ciencias Mención Investigación y Desarrollo de Productos Bioactivos, Instituto de Química de los Recursos Naturales, Universidad de Talca, Talca 3460000, Chile; 3Departamento de Estomatología, Facultad de Ciencias de la Salud, Universidad de Talca, Talca 3460000, Chile; wdonoso@utalca.cl; 4Unidad de Anatomía Patológica, Hospital Regional de Talca, Talca 3460001, Chile; icastillo@ucm.cl; 5Centro Oncológico, Facultad de Medicina, Universidad Católica del Maule, Talca 3466706, Chile; 6Departamento de Bioquímica Clínica e Inmunohematología, Facultad Ciencias de la Salud, Universidad de Talca, Talca 3460000, Chile; rorrego@utalca.cl

**Keywords:** omega-3 derivatives, eicosanoids, liver fibrosis, apoptosis, microsteatosis

## Abstract

Liver fibrosis is a complex process associated to most types of chronic liver disease, which is characterized by a disturbance of hepatic tissue architecture and the excessive accumulation of extracellular matrix. Resolvin E1 (RvE1) is a representative member of the eicosapentaenoic omega-3 lipid derivatives, and is a drug candidate of the growing family of endogenous resolvins. Considering the aforementioned, the main objective of this study was to analyze the hepatoprotective effect of RvE1 in a rat model of liver fibrosis. Male Sprague-Dawley rats received diethylnitrosamine (DEN, 70 mg/mg body weight intraperitoneally (i.p)) as an inductor of liver fibrosis once weekly and RvE1(100 ng/body weight i.p) twice weekly for four weeks. RvE1 suppressed the alterations induced by DEN, normalizing the levels of alanine aminotransferase (ALT), albumin, and lactate dehydrogenase (LDH), and ameliorated DEN injury by decreasing the architecture distortion, inflammatory infiltration, necrotic areas, and microsteatosis. RvE1 also limited DEN-induced proliferation through a decrease in Ki67-positive cells and cyclin D1 protein expression, which is related to an increase of the levels of cleaved caspase-3. Interestingly, we found that RvE1 promotes higher nuclear translocation of nuclear factor κB (NF-κB)p65 than DEN. RvE1 also increased the levels of nuclear the nuclear factor erythroid 2–related factor 2 (Nrf2), but with no antioxidant effect, measured as an increase in glutathione disulfide (GSSG) and a decrease in the ratio of glutathione (GSH)/GSSG. Taken together, these results suggest that RvE1 modulates the fibrogenesis, steatosis, and cell proliferation in a model of DEN induced fibrosis.

## 1. Introduction

Liver fibrosis is a dynamic response to chronic liver injury caused by various agents, such as viruses, alcohol, metabolic and autoimmune disorders [1]. Liver fibrosis is a complex process that involves several cells of the hepatic sinusoid, which is characterized by a disturbance of the architecture and composition of extracellular matrix (ECM) [2]. Under certain circumstances, the acute inflammation can lead to persistent chronic inflammation, which when unresolved may promote organ fibrosis and dysfunction, ultimately leading to cirrhosis and its consequences: portal hypertension, hepatocellular carcinoma (HCC), and liver failure [3]. Hepatic inflammation is a hallmark of the early stage fibrosis, which can progress to extensive fibrosis and cirrhosis [4]. According to the last report from the Centers for Disease Control and prevention (CDC), cirrhosis and chronical liver disease (CLD) are the eleventh cause of death affecting millions of patients worldwide [5]. Globally, the age-standardized incidence rate of cirrhosis and CLD was 20.7 per 100,000 inhabitants, and the estimated incidence of cirrhosis in Europe is 26.0 per 100,000 [6].

The pro-resolving lipid mediators, such as lipoxins and other specialized pro-resolving mediators (SPM), are crucial for the active resolution of inflammatory processes [3]. SPM were coined as lipoxins, resolvins, protectins and maresins [7,8,9,10]. These SPM are potent inhibitors of polymorphonuclear (PMN) leukocyte transendothelial migration and infiltration in vivo [7]. Resolvin E1 (RvE1) is a representative member of the SPM omega-3 lipid autacoids, and is a drug candidate of the growing family of endogenous resolvins [11]. At very low concentrations, RvE1 promotes the resolution of acute inflammation by regulating leukocyte infiltration, increasing the macrophage ingestion of apoptotic neutrophils by macrophage, and enhancing the clearance of phagocytes in the lymph nodes [12], spleen, and heart [13]. Also, RvE1 mediated survival and reinforced the shift toward apoptosis [14]. In relation to CLD, RvE1 reduces the levels of serum indicators of liver fibrosis, such as laminin, hyaluronic acid, pro-collagen type III, and type IV collagen [15]. Also, plasma RvE1 increased in patients undergoing major hepatobiliary resection when EPA was previously administrated (immunonutrition) and the levels of RvE1 were correlated with plasma IL-6 after operation [16]. Considering the previous beneficial results in CLD, the main objective of this study was to analyze the hepatoprotective effect of RvE1 on a rat model of liver fibrosis.

Animal models to investigate the potential hepatoprotective drugs that have a positive impact in liver fibrosis have been well defined and diethylnitrosamine (DEN) is considered one of the most toxic agents that can cause several necroses and consequent fibrosis [17,18]. Also, DEN administration causes excessive deposition of extracellular matrix proteins (collagen) in the rat liver and appears to be appropriate for the study of the early events associated with the development of hepatic fibrosis [19]. Interesting, DEN is a carcinogenic substance that forms DNA adducts through alkylating metabolites that are generated through cytochrome P450 enzymes [20]. Also DEN induces irreversible hepatocellular carcinogenesis through the overexpression of G1/S phase regulatory proteins through the promotion of changes in the expression of cell cycle regulatory factors [21]. Due to the carcinogenic properties, DEN has become a highly attractive experimental model to study liver tumorigenesis [20] and the mechanism of fibrosis associated to cell cycle, inflammation, and hepatitis towards liver carcinogenesis.

## 2. Results

To investigate the efficacy of RvE1 to attenuate the liver fibrosis or reverse it, rats were submitted to chemically induced liver fibrosis, and the impact of RvE1 was studied (growth chart (in grams) is available in Appendix A). Biochemical parameters analysis show that DEN induced liver alterations related to increases in ALT, aspartate transaminase (AST), and LDH values, and reductions of albumin levels (Table 1), with insignifficant alterations in the values of Alkaline phosphatase (ALP) or γ-glutamyl transferase (γGT) in relation to control. The RvE1 administration concomitant to DEN normalized the levels of ALT, albumin and LDH (1.6, 1.7 and 1.7 folds respectively *p* < 0.05) and show a slight decrease of the levels of AST (*p* = 0.1780) and LDH (*p* = 0.2904).

The histological assessment of the livers showed that DEN administration generates architectural disorder, with dilatation of the sinusoid (peliosis), substantial confluent and multiacinar liver necrosis, with moderate to severe portal inflammation and degenerative changes in comparison with control group (Figure 1A). Also, the DEN damage was quantified as cytoarquitecture, inflammation and necrosis by a blinded pathologist (Figure 1B–D). RvE1 administration ameliorated DEN injury (RvE1 + DEN) by decreasing the degree of architecture distortion, inflammatory infiltration, and necrosis areas (2.9-, 1.7-, and 2.3- fold, *p* < 0.0017, *p* < 0.0296, and *p* < 0.0006 respectively) (Figure 1). Next, mitotic activity index was measured. DEN administration generated a doubling of the number of cells that enter in mitosis in relation to the RvE1 (*p* < 0.005, Figure 1E). Masson’s Trichrome and α-smooth muscle Actin (α-SMA) staining analysis (Figure 2A) revealed, as expected, an exaggeration of ECM and a tendency to form fibrotic bridges in the DEN groups when is compared with all the other groups, situation that is reversed by the administration of RvE1, where it is observed a reduced of the fibrotic total area. Fibroblast activity was measured by α-SMA positive area quantification (Figure 2B), showing a 2.2-fold (*p* < 0.0001) increase in DEN group related to control, and RvE1administration depleted this value in a 0.7-fold (*p* < 0.0011). See Appendix A for more histological details.

Pathological analysis of fat droplets deposits (microsteatosis) was measures by Oil Red O staining in the DEN induced fibrosis model (Figure 3A,B). Total lipid area quantified in DEN represent an 9% when is compared with controls groups (CC and RvE1), which shows values less than or near to 1% of the total area of analysis (fields). RvE1 + DEN presented a significative decrease in the total percentage of Oil Red O staining when compared to DEN (65% less staining, *p* < 0.05).

The effects of RvE1 administration on cell proliferation was measured in hepatocytes using the proliferation marker Ki67 (Figure 4A). Whereas few Ki67- positive hepatocytes were observed in control livers (2.1 ± 1.9% per field of view), many were observed in DEN–treated liver (13.4 ± 5.3% per field of view) (Figure 4B) and the RvE1 + DEN evidence a decrease in the number of positive Ki67 hepatocytes (5.2 ± 2.3% per field of view) compared to DEN (*p* < 0.05). In order to determines whether RvE1 has a role on hepatocyte apoptosis in the DEN-induced liver fibrosis, the presence of cleaved-caspase 3 was studied. Figure 4C shows that, in DEN-induced liver fibrosis, there was a higher cleaved-caspase 3 expression compared to control (1.7-fold, *p* < 0.05), while RvE1 group has an increase of 3.44 and 1.9 folds when compared to control and DEN respectively (*p* < 0.05). In concordance with these values, DEN presents an increase in Cyclin D1 protein expression (3.3-fold, *p* < 0.05), while RvE1 did not increase the levels of Cyclin D1 compared to controls (0.4397 ± 0.1658 vs 0.5066 ± 0.1115, respectively) (Figure 4D). When the B-cell lymphoma 2 (Bcl-2) protein, an anti-apoptotic protein, was determined (Figure 4E) it was observed that DEN and RvE1 + DEN decreased by 1.5-fold and 1.7-fold compared to control, respectively. RvE1 do not show any statistic variations respects to DEN in Bcl-2 values (*p* = 0.666). Next, we evaluated the activity of the transcription factors Nrf2 and NF-κBp65, as their translocation to the nucleus (Figure 5A–D). As expected, DEN promotes the nuclear translocation of NF-κBp65 with a diminution in cytoplasmic protein presence (a median of 0.7663 ± 0.2303 to 1.076 ± 0.1203, from cytoplasm to nucleus) compared to control group. Interestingly, the supplement of RvE1 promotes higher nuclear translocation of NF-κBp65 compared to control or DEN (2.7-fold and 2.1-fold respectively, *p* < 0.05). Since these transcription factors are targets of the cytokines, we evaluated the serum levels of tumor necrosis factor (TNF)-α and interleukin (IL)-10 (Figure 5E,F). TNF-α is increased 3.1- and 1-fold in DEN and RvE1 + DEN relative to control, and with non-statistically differences among them (2.066 ± 1.114 and 2.137 ± 0.4848 for DEN and RvE1 + DEN respectively). Further, IL-10 did not show any significant variation among the groups. Then, we analyzed the nuclear translocation of Nrf2 (Figure 5C,D), where DEN groups show a lost signal in nuclear samples compared to all the other groups (*p* < 0.05). Nrf2 was incremented in the groups RvE1 and RvE1 + DEN (1.7- and 1.4-fold) with respect to control. Because Nrf2 is a regulator of oxidative balance, GSH, GSSG, and GSH/GSSG ratio were measured (Figure 5G–I). We observed that DEN group presents a reduction in GSH levels compared to control and was restored in the RvE1-treated groups, while DEN group not present an increase in the levels of GSSG nor GSH/GSSG (*p* > 0.05), probably due to a loss of functional tissue. RvE1 shows an increase in GSSG (1.6- and 1.3-fold respect to control and DEN respectively, *p* < 0.05), and a decrease in the ratio GSH/GSSG (22% and 10% regard to control and DEN respectively).

## 3. Discussion

The CLD is characterized by fibrosis, when fibrosis is self-limited, resulting in a balanced, protective, reparative, and non-tissue injury response. When the regulatory response related to fibrosis become chronic and there is a dominance of the repetition of the injury, the benefits of the fibrosis balance is lost. The end-stage of chronic liver fibrosis results in cirrhosis and is a limiting factor for the development of HCC. There is important to mention that the only therapeutic approved approaches to removal the injury related to the describes CLD is the liver trasplantation [22,23]. Because the scarcity of donors is a serious limitation, new therapeutic approaches are urgently required and antifibrotic therapies. The present study aimed to investigate whether RvE1 could reduce liver fibrosis and stimulate the resolution of the damaged fibrotic liver.

There is animal models and human interventions that showed the beneficial effect of the EPA and DHA administration on CLD, in particular when these fatty acids are administrated improved the biochemical parameters, inflammation, ameliorates fibrosis, fatty acid cumulation, improving esteatosis [24,25,26]. Interestingly, when EPA is compared with DHA in a model of non-alcoholic fatty acid liver disease (NAFLD), EPA is better in controlling hepatic triglyceride cumulation and steatosis, but is DHA the fatty acids who is in charge of controlling the inflammation and oxidative stress, and both are important in the control of fibrosis, acting sinergistically [27]. The above is interesting, since resolvins are derived from EPA or DHA. Thus, here, we present the results obtained from the RvE1, an EPA derived bioactive SPM. In concordance with the above, Gonzalez-Periz et al. reported that the protective effect of omega-3 fatty acids observed in their model (ex-vivo and in vivo) was mediated by protectin D1 and RvE1. RvE1 administration (1.2 ng/g) conferred protection against hepatic steatosis, decrease of liver injury serum markers, and macrophage infiltration [28]. Qiu et al. demonstrated that RvE1 (100 ng for 70 days) improves the liver fibrosis caused by the infection of *Schistosoma japonicum* to mice. RvE1reduces the levels of transaminases, TNF-α and INF-γ, and lower the levels of fibrotic markers such as laminin, hyaluronic acids and pro-collagen III among other parameters [15]. In our study, RvE1 normalized liver parameters and demonstrated potent antifibrotic activities. Also, recently, it was shown that the RvE1 concentration was decreased during NAFLD progression [29], and this could be related with the increase in the omega-6/omega-3 ratio which is associated with lipogenesis and the lipid oxidation promotion of NAFLD [30].

Even though the model of DEN is not the ideal system to study the phenomena of liver steatosis, it has been described that it is a support model to study metabolic and nutritional changes in the fatty acids deposits [31,32,33]. Histologically, the liver is considered steatotic when >5% of hepatocytes in a tissue section stained with H&E contain macrovesicular steatosis [34,35,36]. Interestingly, H&E often underestimate steatosis by the difficulties in detecting lipid microvesicles after paraffin embedding, and underestimation of the degree of fatty changes [37,38]. On the other hand, Oil-Red O is fast and sensitive and improve the assessment of liver steatosis [38,39]. The fact that RvE1 reduced the levels of fatty droplets in our experiments could be interesting to introduce future studies of SPM in the context of steatosis-related liver fibrosis and even HCC. The aforementioned becomes more interesting if we consider that Kuang et al. demonstrated that RvE1 and RvD1 can reverse the development of liver cancer cells in a long-term concavalin-A induced injury [40].

The steatotic hepatocyte is associated with dysregulated lipolysis resulting in excessive delivery of fatty acids to the liver, de novo lipogenesis, impaired post-receptor signaling by insulin (i.e., insulin resistance) [41]. It has been proposed that the downregulation of steroyl-CoA response element binding protein-1c (SREBP-1c), a de novo lipogenesis related transcriptional regulator could improves the non-alcoholic stetatohepatitis (NASH) [41,42]. EPA administration decrease the expression of SREBP-2 in NAFLD [43] and SREBP-1c in hepatocyte cell culture [44,45]. Among resolvins, only RvD1 has been assayed by their role on SREBP-1 [46], here the authors found that RvD1 protect endoplasmic reticulum stress in HepG2 cells, this protection was related to a decrease in caspase activity and SREBP-1 expression [46]. Previously, Neuhofer et al., demonstrated that the deficiency of protectin D1 and 17-HDHA (the precursor of RvD1) is linked to the development and perpetuation of obesity-driven adipose tissue inflammation that promotes type 2 diabetes, and this may be corrected by the addition of 17-HDHA resulting in increased levels of peroxisome proliferator-activated (PPAR)γ, PPARα and adiponect [47]. The mentioned factors are involved with the regulation of fatty acids metabolism, insulin sensitivity, prevention of hepatic steatosis, and interestingly they contribute to the anti-inflammatory milieu in NASH/NALFD [48,49]. As previously was reported by Gonzalez-Periz, RvE1 protects against liver steatosis by improving insulin tolerance and induction of PPARγ and adiponectin [28]. More recently, Barden et al., found that RvE1 was produced in large amounts in patients with metabolic syndrome who have lost weight [50]. On the other hand, Maciejewska analyzed the relationship of eicosanoid with steatosis progression in animal model, finding a moderate correlation of RvE1 decreasing with NAFLD progression, but in liver tissue the association is weak and results difficult to associated to results observed in serum with the tissue liver observed [29]. The previous results suggest that this SPM is sensitive to fat changes in the tissues. The fact that RvE1 administration depleted the microsteatotic deposits in our model (fibrosis) confirms the role of lipid metabolism independently of the model of study, and it will be of interest to study the role of RvE1 on SREBPs and PPARs transcription factors.

In CLD, it is commonly believed that simple steatosis is the benign, while NASH is progressive with marked increase in liver-related mortality due to cirrhosis or HCC and HCC is common in patients with fibrotic NASH and NASH related cirrhosis [51]. Also, metabolic factors, such as diabetes and obesity, can contribute to HCC [52]. It has been confirmed that the production of reactive oxygen species (ROS) and inflammatory infiltration promotes cellular proliferation and HCC induction [53]. Liver cell proliferation and its relation with tumor growth can be evaluated by Ki67 expression [54,55]. Previously, El-Kebir et al. confirmed that RvE1 enhances phagocytosis and induces neutrophil apoptosis in isolated neutrophils, mediated by a caspase-8 and caspase-3 activation [56]. In our model, we described that RvE1 decreased the mitotic activity index, with Ki67 positive cell staining enhancing the activity of caspase-3 and limiting the expression of cyclin D1 and BCL-2, all of these related to a potential inhibition of damaged tissue proliferation.

Previously, it has been demonstrated that nanomolar concentration of RvE1 can blocks PMN infiltration and attenuated dendritic cell migration, activity that is related to his bound to ChemR23 a G-protein-coupled receptor [57]. The Chem23 selectively union blocks TNF-α signaling, also RvE1 binds to leukotriene B4 (LTB_4_) receptor on human PMN and the union to these receptor apparently mediated the counter regulation of RvE1-ChemR23 to promote acute inflammation resolution [58]. The up-regulation of TNF-α is associated with the enhancement of DNA binding activity of NF-κB activity which is downregulated and modulated by EPA and DHA administration, with the consequence of anti-inflammatory and ROS reduction [59,60]. Also, RvD1 and RvE1 significantly downregulate CD^4+^ and CD^8+^ liver infiltration and the related injury, throughout inhibition of pro-inflammatory cytokine discharge and NF-κB/AP-1 nuclear activity [40]. Based in this information, it is expected that the administration of RvE1 downregulates the levels of TNF-α related to an enhance of nuclear translocation of NF-κB, but in our model, we do not observe molecular anti-inflammatory behavior, we only detected a decrease in the inflammatory infiltration in the liver tissue. The fact that neither TNF-α nor IL-10 show changes in our model is presumably related to the crosstalk among NF-κB and oxidative stress [61]. We presume that higher doses of RvE1 can block the movement to NF-κB at the nucleus and promote the expected anti-inflammatory phenotype. In relation with this, it is well described that the activation of Nrf2 antagonizes the inflammatory pathways of transforming growth factor (TGF)-β1 and NF-κB, and plays a cytoprotective role in cell damage [62], where the disruption or loss of Nrf2 signaling causes enhanced susceptibility not only to oxidative and electrophilic stresses, but also to inflammatory tissue injuries [63]. Nrf2 regulates apoptosis and fibrotic process through activation of antiapoptotic Bcl-2 protein [4]. It should be mentioned that exists a crosstalk between Nrf2 and NF-κB, where the absence of the first can exacerbate cytokine production, whereas NF-κB can modulate Nrf2 transcription and activity. Both Nrf2 and NF-κB are regulated by redox sensitive factors, but NF-κB is more readily activated in oxidative environments, another interesting fact is that Nrf2 contains several κB sites in its proximal promoter, which are subject to be binding p65, suggesting that high activation of NF-κB signaling inhibits/repress Nrf2-ARE pathway [64,65,66]. This can explain why the enhancement of nuclear Nrf2 was not reflected in an improvement in ROS an inflammatory status. Finally, it is necessary to mention that Pohl et al. did not find positive changes in liver NASH or fibrotic changes when RvE1 was administrated (1.2 ng/g body for four days) [67].

The data reported here support the role of RvE1 as hepatoprotective by controlling the fibrogenesis, steatosis, and cell proliferation in a model of DEN induced fibrosis. Nevertheless, it has to be considered that some of the experiment results does not resemble the expected literature. Thus, further studies, eventually using higher doses of RvE1, decreasing the time between the doses, or increasing the time of exposition (weeks), could show the described anti-inflammatory effect. Also, we propose to compare it with other resolvins and SPMs should be assayed to clarify the activity on liver fibrosis.

## 4. Materials and Methods

### 4.1. Animals

Male Sprague-Dawley rats (70–110 g) were obtained from Bioterio Central, Dirección de Investigación, Universidad de Talca, Chile. Animals were allowed free access to food (Champion S.A., Santiago, Chile) and water, and were housed in a temperature-controlled room on a 15 h light/dark cycle. All animals received humane care in compliance with the University’s guidelines and the ethics statement, experimental animal protocol and animal procedures in this project was complied with the “Guide for the Care and Use of Laboratory Animals” (National Academy of Sciences, NIH Publication 6-23, revised 1985). All experiments were approved by the Bioethical Committee (CIECUAL), Folio number 2016-06B-C, Dirección de Investigación, Universidad de Talca. All the experiments with animals proposed in this project was made under the supervision of a veterinarian expert.

### 4.2. Model of Liver Fibrosis

To induce liver fibrosis induction, DEN (Cat No. 73861, Sigma-Aldrich, Merck KGaA, Darmstadt, Germany) was administrated intraperitoneally (i.p) at doses of 70 mg/g body weight (in 0.9% NaCl), once a week for a period of 4 weeks according to the model of DEN-induced liver fibrosis [18]. RvE1 (Cat No. 10007848, Cayman chemical, Ann Arbor, MI, USA) was administrated i.p twice a week for the same 4 weeks of DEN treatment (100 ng/body weight [15]). Animals were randomly assigned to one of the following groups: (i) vehicle DEN + 0.025% ethanol in 0.9% NaCl (vehicle of RvE1) (group control-control/CC); (ii) DEN + vehicle RvE1 (group DEN); (iii) vehicle DEN + RvE1 (group RvE1) and (iv) DEN + RvE1. At the end of 4 weeks, the animals were anaesthetized with (1 mL/kg) of zolazepam chlorhydrate (25 mg/mL) and tiletamine chlorhydrate (25 mg/mL) (Zoletil 50™; Virbac S/A, Carros, France). Blood and liver samples was taken from the medial lobes for experimental analysis. Animal number per experimental group: *n* = 6–9.

### 4.3. Measurement of Biochemical Parameters

ALT, AST, Albumin, ALP, LDH, γ-GT were measured using specific diagnostic kit (LiquidColor Human™, Wiesbaden, Germany). To control the measurements, adequate 2-level controls, normal and pathological, were used. ELISA kits were used for assessment of serum levels (pg/mL) of TNF-α and IL-10 (Thermo Fischer Scientific, Rockford, IL, USA).

### 4.4. Liver Glutathione Assay

Livers were perfused in situ with washing solution (159 mM KCl and 5mM Tris, pH 7.4) to remove blood and measured liver content of reduced (GSH) and oxidized (GSSG) were measured in deproteinated tissue homogenates samples using a colorimetric Glutathione assay Kit (Cat No. 703002, Cayman Chemicals, Ann Arbor, MI, USA).

### 4.5. Histopathological Staining

For Hematoxylin & Eosin (H&E) and Masson´s Trichrome stain (Merck KGaA, Darmstadt, Germany), a third part of the lobes were fixed in 10% buffered formalin, embedded in paraffin, and sectioned in 5 μm, before being stained and analyzed for morphology, cell infiltration, collagen deposition, and mitotic index activity (MAI). Analysis was performed blind by a pathologist (I.C) according to the procedure reported previously by our group [68], following the Koroukian score [69] and Goodman’s adapted Ishack score [70,71]. For Oil Red O (0.5%) (Sigma-Aldrich, Merck KGaA, Darmstadt, Germany), fresh liver samples were frozen and cut in cryostat microtome (5 μm thickness, −20 °C). Gill´s III modifies Haematoxylin (Merck) was used as counterstain. Mitotic activity index (MAI) was analysed according to Al-Janabi [72]. All the histological analyses were performed in a Nikon Eclipse 50i Optic microscope (Nikon, Tokyo, Japan), while the posterior analysis and photograph were developed in Micrometrics SE PremiumTM software (Opticstar, Manchester, UK).

### 4.6. Inmunohistochemitry Staining

Liver sections were immunostained for Ki67 (1/300, mouse monoclonal, Merck Millipore, Burlington, MA, USA) and α-SMA (1/25, mouse monoclonal, Leica biosystem, Weszlar, Germany). Immunohistochemistry was performed on sequential paraffin liver sections (4-μm thick) incubated with specific monoclonal antibodies. Briefly, after microwave antigen retrieval with 0.01 mol/L citrate buffer, pH 6.0, primary antibodies were labeled using VectorStain^®^ Elite^®^ ABC kit peroxidase HRP (Cat No. PK6200, Vector Laboratories, Maravai LifeSciences, CA, USA). Biotinilated secondary antibodies directed against mouse antigens and visualized by Avidin plus Biotinylated ABC reagents (Vector). Negative controls were performed by replacing the respective primary antibodies by isotype and concentrations matched irrelevant antibody (Human oral pyogenic granuloma). The images were obtained using the Leica Microsystems Limited Software v 3.4.0 (Stereo and Macroscope Systems, Heerbrugg, Switzerland) and the images were quantified by ImageJ software v 2.0.0-rc-69/1.53a (NIH, Bethesda, MD, USA, https://imagej.nih.gov/ij/docs/examples/stained-sections/index.html).

### 4.7. Western Blot Analysis

Cytoplasmic and nuclear samples were obtained from the frozen hepatic tissue samples (200 mg) from the adapted protocol of Deryckere and Gannon [73] and described in Soto et al. [68]. Briefly, frozen liver was homogenized and suspended in buffer solution pH 7.9 (10 mM 4-(2-hydroxyethyl)-1-piperazineethanesulfonic acid (HEPES), 1 mM ethylenediaminetetraacetic (EDTA), 0.6% nonidet P (NP)-40, 150 mM NaCl, and 0.5 mM phenyl methyl sulphonyl fluoride (PMSF)), followed by centrifugation at 3.020× *g* for 15 s at 4 °C. The supernatant corresponds to cytoplasmic fractions. The precipitate was resuspended in 200 μL of nuclear buffer solution pH 7.9 (20 mM HEPES, 0.2 mM EDTA, 25% glycerol, 420 mM NaCl, 1.2 mM MgCl2, 0.5 mM di-thio threitol DTT, 0.5 mM PMSF, 2 mM benzamidine, an Pierce protease inhibitors cocktail mini tablest^®^ (Pierce, Thermo Fischer Scientific, Rockford, IL, USA)), followed by centrifugation at 13,000× *g* for 60 s, and the supernatant was incubated for 20 min in ice. Then, the supernatant was centrifugated at 13,000× *g* for 30 s at 4 °C to eliminate nuclear debris (precipitate). Cytoplasmic and nuclear protein fractions (50 μg) were separated on 12% polyacrylamide gels using sodium dodecyl sulfate polyacrylamide gel electrophoresis (SDS-PAGE by using Mini-Protean^®^ and Protean II^®^ systems (Bio-Rad Laboratories, Hercules, CA, USA)) and transferred to nitrocellulose membranes, which were blocked for 1 h at 22 °C with TBS containing 5% skim milk. The blots were washed with TBS containing 0.1% Tween 20, hybridized with rabbit polyclonal primary antibodies, for either Nrf2 (1:500), NF-κBp65 (1:1000), active caspase-3 (1:200), and histone H1 (1:350) as a nuclear housekeeping protein. Mouse monoclonal primary antibodies used were for cyclin D1 (1:1000), BCL-2 (1:500), β-actin (1:2000), and GAPDH (1:2000), with these last two used as a cytoplasmic housekeeping protein (Nrf2, NF-κB, active caspase-3, histone H1, cyclin D1, GAPDH and secondary antibodies were purchased from Merck Millipore, Burlington, MA, USA; BCL-2 from Thermo Fischer Scientific, Rockford, IL, USA; and β-actin from Santa Cruz Biotechnology, Dallas, TX, USA). The antibodies were incubated overnight at 4 °C. After extensive washing, the antigen antibody complexes were detected using horseradish peroxidase-labeled goat anti-rabbit IgG/anti-mouse or rabbit, and the protein was detected with the protein detection kit using Amersham enhanced chemioluminicense (ECL) Prime Western Blotting Detection Reagent (General Electric Healthcare, Hammersmith, UK). The chemiluminescent signals were analyzed in the Omega Lum™ System (Aplegen, San Francisco, CA, USA), and the quantification of luminescent images was made in ImageJ (NIH). For all protocolos general reagents and molecular standars were purchased from Winlker LTDA (Curicó, Maule, Chile).

### 4.8. Statistical Analysis

Values are presented as the mean ± standard deviation (SD). The number of samples are indicated in each figure. Student´s *t*-test for unpaired data or one-way analysis of variance (ANOVA) with the Tukey’s test as a post-hoc test was used to assess differences between means of the different groups. For non-parametric data, the Kruskal–Wallis or Mann–Whitney test were used. A *p*-value of less than 0.05 was considered significant. The analyses were performed using the GraphPad Prism 6.0 software (GraphPad software, Inc. San Diego, CA, USA).

## Figures and Tables

**Figure 1 ijms-21-08827-f001:**
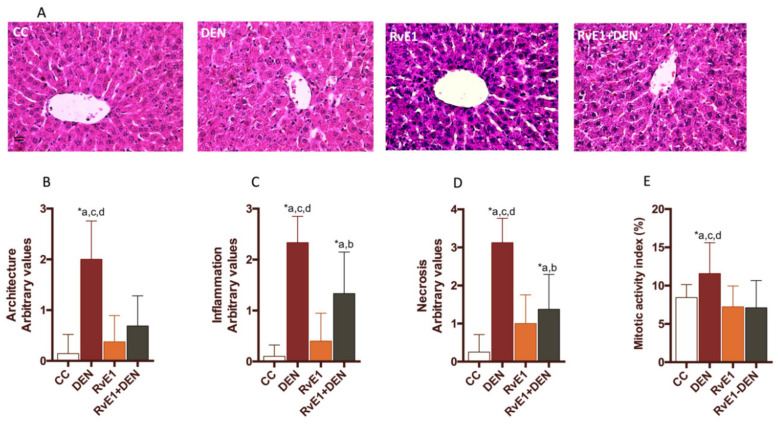
Effect of RvE1 on liver morphology. (**A**) Representative liver sections stained with hematoxylin-eosin. Scores of livers sections were graphed for (**B**) architecture, (**C**) inflammation, (**D**) necrosis and (**E**) mitotic activity index (MAI) measured as a percentage of a whole section. At least 20 fields for every sample were analyzed at 400× magnification. *n* = 6 to 9 animals per experimental group. Asterisk indicates *p* < 0.05, and the letters identify the experiments that are compared and present this statistical difference. a = control; b = DEN; c = RvE1 and d = RvE1 + DEN.

**Figure 2 ijms-21-08827-f002:**
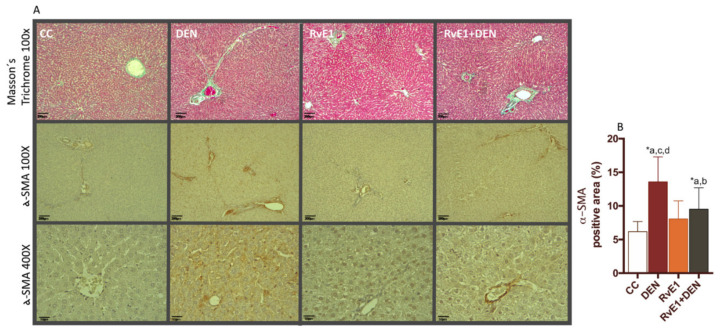
Effect of RvE1 on matrix deposit. (**A**) Representative histopathological microphotography of liver. Upper panel: Representative Masson´s trichrome. Middle panel: Representative images of α-SMA staining at 100×. Lowe panel: Representative images of α-SMA staining at 400×. (**B**) Quantification of α-SMA positive areas. At least 20 fields for every sample were analyzed at 400× magnification. *n* = 6 animals per experimental group. Asterisk indicates *p* < 0.05, and the letters identify the experiments that are compared and present this statistical difference.

**Figure 3 ijms-21-08827-f003:**
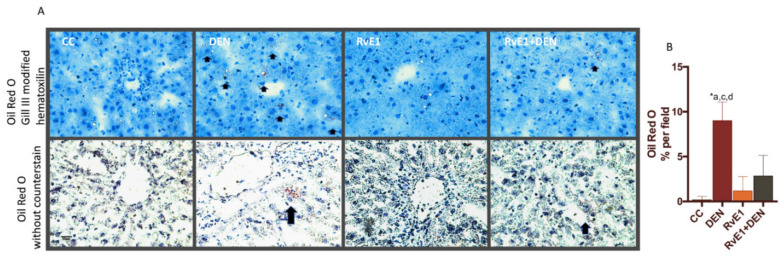
Effect of RvE1 on lipid deposit. (**A**) Representative histopathological microphotography of liver. Upper panel: Oil Red O plus counterstain Gill III modified hematoxylin. Lower Panel: Oil Red O without counterstain. (**B**) Quantification of Oil Red O percentage per field. Magnification 400×. At least 20 fields for every sample were analyzed. *n* = 6 animals per experimental group. Asterisk indicates *p* < 0.05, and the letters identify the experiments that are compared and present this statistical difference. Arrow indicates the high areas of Oil Red O cumulation.

**Figure 4 ijms-21-08827-f004:**
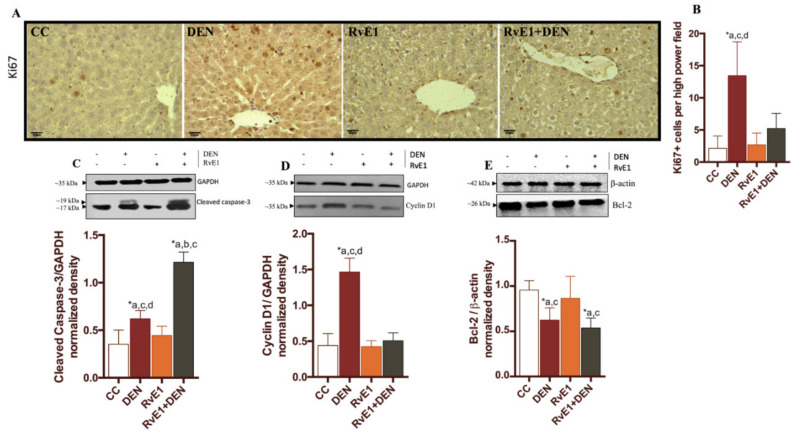
Effect of RvE1 on cell cycle and apoptosis. (**A**) Representative histopathological microphotography tissue Ki67 at 400× magnification. (**B**) Quantification of Ki67 positive cell. Western blot analysis of (**C**) Cleaved caspase-3, (**D**) Cyclin D1 and (**E**) BCL-2. The levels were normalized to GAPDH or β-actin as housekeeping. *n* = 6–9 rats per experimental group. Asterisk indicates *p* < 0.05, and the letters identify the experiments that are compared and present this statistical difference.

**Figure 5 ijms-21-08827-f005:**
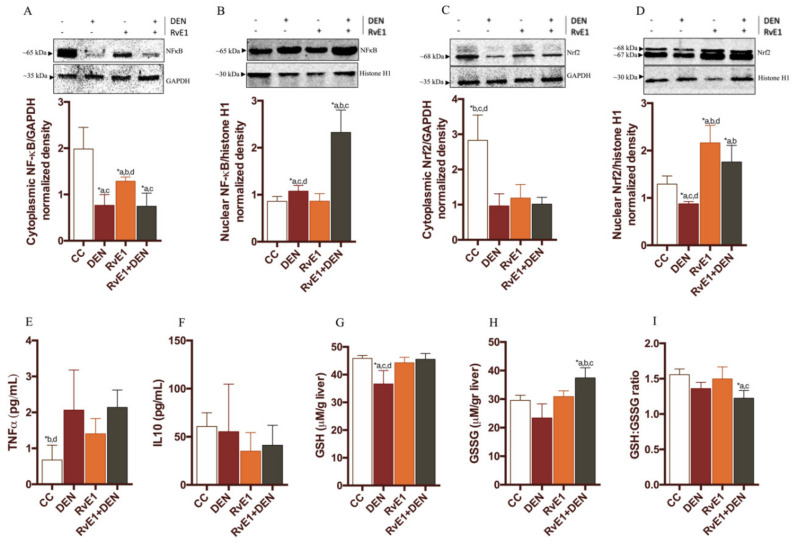
Effect of RvE1 on tissue levels of NF-κB and Nrf2, and molecules targets. Western blot analysis of (**A**) cytoplasmic and (**B**) nuclear NF-κB; and (**C**) cytoplasmic and (**D**) nuclear Nrf2. The cytoplasmic levels were normalized to GAPDH as housekeeping and nuclear levels were normalized to histone H1 as housekeeping. (**E**,**F**) serum levels of TNF-α and IL-10. (**G**–**I**) tissue levels of GSH, GSSG and GSH:GSSG ratio. *n* = 6–9 rats per experimental group. Asterisk indicates *p* < 0.05, and the letters identify the experiments that are compared and present this statistical difference.

**Table 1 ijms-21-08827-t001:** Serum clinical values of the Sprague-Dawley rats. Data are expressed as mean ± SD. The * means *p* < 0.005.

Groups			Parameters			
	ALT (UI/L)	AST (UI/L)	ALP (UI/L)	Albumin (g/dL)	γGT (UI/L)	LDH (UI/L)
**CC (a)**	52.76 ± 26.10	112.5 ± 26.41	569.5 ± 114.8	2.381 ± 0.45	1.857 ± 0.38	358.3 ± 203.0
**DEN (b)**	123.3 ± 26.52 *^b,c,d^	199.7 ± 54.94 *^a^	833.2 ± 145.0	1.681 ± 0.16 *^a,c,d^	2.000 ± 0.0	1058 ± 710.0 *^a,c^
**RvE1(c)**	79.89 ± 25.89	164.2 ± 33.64	553.9 ± 116.8	2.387 ± 0.48	1.750 ± 0.5	269.5 ± 197.5
**RvE1 + DEN (d)**	77.52 ± 29.5	166.3 ± 52.12 *^a^	653.6 ± 86.69	2.912 ± 0.55	2.000 ± 0.0	635.7 ± 185.8

ALT alanine aminotransferase, AST aspartate aminotransferase, ALP alkaline phosphatase, γGT gamma glutamil transferase, LDH lactate dehydrogenase.

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
