# Peer review of "Pro-Resolving Lipid Mediator Resolvin E1 Mitigates the Progress of Diethylnitrosamine-Induced Liver Fibrosis in Sprague-Dawley Rats by Attenuating Fibrogenesis and Restricting Proliferation"

_ijms, 2020, doi:10.3390/ijms21228827_

Round 1
Reviewer 1 Report
Authors investigate the protective effect of resolving E1 in a diethylnitrosamine model of liver injury. They find that resolving E1 has a protective effect (in line with results in other experimental models of liver injury). Surprisingly they observed no modulation of inflammatory cytokines and oxidative stress markers.
In general the article is written for peoples expert in liver injury models but more explicative approach is required for a generalist journal. Several reports in the literature indicates a positive effect of resolvin E1 and D1 in hepatotoxicity models (few described in the introduction, some are cited in the discussion).
Although an overall scientific soundness is present, some perplexities remains:
- How did the authors decide the doses and schedules of resolving E administration? They invoke different dosing to justify discrepancies with other studies.
- the paper heavily relies on histopathological scoring and grading. What are the steps undertaken to avoid researcher bias (independent pathologist evaluation, blind scoring, random fielding …)?
- The last part with the study of the mechanisms is less convincing.
- Western blot in Fig. 4C are very poor and presented without molecular weights. No control of cytoplasm/nucleus fractionation is presented (H1 in cytoplasm and GAPDH in nucleus) and data are reported as “relative to GAPDH” or “relative to H1”. Is not stated how they were normalized among the different samples (the experiment involves 9 animals x 4 groups = 36 samples that cannot be run in a single gel).
- Absence of modulation of inflammatory TNFa and IL10 suggest a wrong timing and is conflicting with reported reduction of inflammatory infiltration. Unfortunately, the diethylnitrosamine model of liver injury and the pathogenic mechanisms involved are not described in the introduction.
- Absence of alterations in oxidative stress balance is also puzzling, maybe other markers such as lipid peroxidation may help?
Author Response
Dear Reviewer 1
My colleagues and I have taken into consideration the recommendations made to our manuscript entitled “Pro-resolving lipid mediator Resolvin E1 mitigates the progress of diethylnitrosamine-induced liver fibrosis in Sprague-Dawley rats by attenuating fibrogenesis and restricting proliferation” (ijms-870629). We have made the changes accordingly. We very much appreciate all the reviewer’s suggestions.
|
Comments and suggestions |
Reply |
|
There was a mistake in the form of the data of RvE1 dose was described and it was corrected in the manuscript (Line 309-3011). The doses were elected according to Qiu et al., 2014 (a rodent model of fibrosis) reference 15. The time of administration was related to the DEN model (4 weeks) according to Kim et al., 2016.,reference 18. The times at weeks (twice) was decided in accordance with the directrices of Bioethical committee. |
|
The histopathological analysis was made by a pathologist at Hospital Regional de Talca, a scientist who is not related to the sampling of animals in our laboratory (independent). He analyzed the samples blinded. The liver sections were randomly codified (the corresponding code was revealed at the end of the study) |
|
We have improved the images 4 and 5 related to Western blots: added molecular weight, changed images for a better visualization. Also, it was corrected the error “relative” to the normalization to housekeeping realized (normalized experimental signal =observed experimental signal/ normalization factor*). All the samples for Western blot were assayed by triplicate.
Figure 4, the western blot corresponds to total protein extraction. Figure 5, the Western blot corresponds to differential protein extraction by differential centrifugation according to Deryckere and Gannon (reference 74 y 69) and we use all the internal controls to ensure no contamination or other troubles with the extraction(also, we have used this method for techniques as sensitive as EMSA doi:10.1371/journal.pone.0028502 and 10.3109/10715762.2010.485995)
* normalization factor= observed signal housekeeping for each lane/highest observed signal of housekeeping on the blot The abundance of the housekeeping was constant across our experimental conditions |
|
2. Absence of modulation of inflammatory TNFa and IL10 suggest a wrong timing and is conflicting with reported reduction of inflammatory infiltration.
Unfortunately, the diethylnitrosamine model of liver injury and the pathogenic mechanisms involved are not described in the introduction.
|
We acknowledge that this result is disconcerting. As the reviewer suggest maybe more time of exposition should be considered for to improve the model of study.
We added a description of DEN mechanism in introduction and in discussion. |
|
3. Absence of alterations in oxidative stress balance is also puzzling, maybe other markers such as lipid peroxidation may help?
|
Unfortunately, due to Covid-19 pandemic, our laboratory and university is under lockdown and we cannot afford to perform new experiments. |
Reviewer 2 Report
In this study, Rodriguez et al. seed therapeutic effects of RvE1 on liver damage using DEN rat models. Some previous studies showed therapeutic effects of RvE1 but some studies showed controversial data, such as Pohl et al., as the authors mention in discussion. Therefore, it is important to design models and experiments carefully and provide various evidence to confirm the effects of RvE1. This study’s model is questionable for some experiments. Data provided here lack sufficient evidence as well as objective data. Some data are inconsistent between images and graphs. Overall, this manuscript is poor and not worth for readers. Specific points that need to be addressed before publication are listed below.
- Figure 1 is merely H&E staining and bottom graphs (Figure 1B-E) are subjective data. The authors need to provide objective data to support those subjective evidence. I cannot see much differences especially between DEN and RvE1+DEN in Figure 1A, but the authors show significant differences in bottom graphs. I cannot believe Figures 1B-E and data should be provided by qPCR or commercially available kits.
- In Figure 2A, αSMA staining, RvE1+DEN looks more staining than DEN, but Figure 2B shows that DEN has more positive areas than RvE1+DEN. I cannot understand how the authors calculated and made graphs in this study. Images in Figure A are inconsistent between Masson’s and αSMA and this does not make sense. Something might be wrong with procedures.
- DEN causes cancer in the liver so DEN treatments are used to induce HCC in rodents. High fat diet or other diets are used to induce liver steatosis, and generally DEN is not used for steatosis models. In Figure 3, DEN shows limited droplets and under 10% stained areas are generally not recognized as steatosis or abnormal lipid deposit. DEN treatments are not good models to see effects of RvE1 on lipid deposit and Figure 3 does not prove anything.
- In Figure 4C, control group clearly shows higher band intensity for cleaved caspase 3 than DEN, but in Figure 4D, DEN shows higher expression than control. Same problem for cyclin D1 and Bcl2. Figure 5 also has inconsistency between images (Figure 5A) and graphs (Figure 5B-E). I do not understand how the authors produced these graphs and I cannot believe all of them. Data analysis was poorly performed in this study and it damages the quality of this manuscript significantly.
Author Response
Dear Reviewer 2
My colleagues and I have taken into consideration the recommendations made to our manuscript entitled “Pro-resolving lipid mediator Resolvin E1 mitigates the progress of diethylnitrosamine-induced liver fibrosis in Sprague-Dawley rats by attenuating fibrogenesis and restricting proliferation” (ijms-870629). We have made the changes accordingly. We very much appreciate all the reviewer’s suggestions
|
Comments and suggestions |
Reply |
|
· Figure 1 is merely H&E staining and bottom graphs (Figure 1B-E) are subjective data. The authors need to provide objective data to support those subjective evidence. I cannot see much differences especially between DEN and RvE1+DEN in Figure 1A, but the authors show significant differences in bottom graphs. · · I cannot believe Figures 1B-E and data should be provided by qPCR or commercially available kits.
|
Figure 1 was improved for a better visualization. Figure 1B-E are quantification of a pool of microphotography’s using reported criteria (references 70, 71 y 72). The analysis was made by an independent blinded pathologist (From Hospital Regional de Talca, Chile).
It would be interesting to evaluate necrosis or even necroptosis for another techniques, but as explained previously our facilities are under lockdown due to Covid-19 and we are not authorized to enter to the laboratories for to perform new experiments. |
|
· In Figure 2A, αSMA staining, RvE1+DEN looks more staining than DEN, but Figure 2B shows that DEN has more positive areas than RvE1+DEN. I cannot understand how the authors calculated and made graphs in this study. Images in Figure A are inconsistent between Masson’s and αSMA and this does not make sense. Something might be wrong with procedures.
|
Pictures were changed to improve visualization. Each set of microphotographys (columns) of Masson and a-SMA correspond to the same animal (liver) (same sample for control, same for DEN, same for RvE1 and same for RvE1+DEN) with the intention of allowing a better comparison between the techniques. The quantification of a-SMA was described in line 356-359.
|
|
· DEN causes cancer in the liver so DEN treatments are used to induce HCC in rodents. High fat diet or other diets are used to induce liver steatosis, and generally DEN is not used for steatosis models. In Figure 3, DEN shows limited droplets and under 10% stained areas are generally not recognized as steatosis or abnormal lipid deposit. DEN treatments are not good models to see effects of RvE1 on lipid deposit and Figure 3 does not prove anything.
|
As the Reviewer describes, DEN effectively is not the best model to evaluate lipid metabolism, and it was not the main objective of our work. We corrected and improved discussion related to the analysis of percentage of fatty droplets, steatosis and the potential activities of RvE1 in this context. Also, we improved the description of the DEN model and its limitations in the context of this experiment. Line 198-210 and 221-236.
|
|
· In Figure 4C, control group clearly shows higher band intensity for cleaved caspase 3 than DEN, but in Figure 4D, DEN shows higher expression than control. Same problem for cyclin D1 and Bcl2. Figure 5 also has inconsistency between images (Figure 5A) and graphs (Figure 5B-E). I do not understand how the authors produced these graphs and I cannot believe all of them. Data analysis was poorly performed in this study and it damages the quality of this manuscript significantly.
|
We improved the Western blots of images 4 and 5: added molecular weight, changed images for a better visualization. Also, we corrected the error “relative” to the normalization to housekeeping realized. |
Reviewer 3 Report
In this manuscript, Maria Jose Rodriguez et al showed that RvE1 modulates the fibrogenesis, steatosis and cell proliferation in a model of DEN induced fibrosis. The manuscript could be further strengthened with revision denoted below.
1. There are some places that incorrectly and inaccurately write down the manuscript such as "Figure 1A" or "Fig. 4A", Line number 308 "4.7 Western blot analysis" and Line number 337 "4.7 Statistical Analysis" (Same number). Many of places need to be correct. Authors need to pay attention to check of this manuscript.
2. Please describe more detail about Figure 1A otherwise it is difficult to distinguish between the pictures.
3. In Figure 4C, it is hard to say that the cleaved caspase 3 expression was different between control and DEN group because the expression of cleaved caspase 3 was too high in control group.
4. In figure 4F, Bcl-2 expression looks not different between DEN only and RvE1 samples. Based on the data (quantified with beta-actin and Bcl-2), DEN+RvE1 sample compare to RvE1 only.
4. In Figure 5A, I strongly suggest that authors have to reloading samples on the same gel and need to show the same blot eventhough autors showed the GAPDH and Histone H1. Otherwise, it is hard to say that the extraction went well.
Author Response
My colleagues and I have taken into consideration the recommendations made to our manuscript entitled “Pro-resolving lipid mediator Resolvin E1 mitigates the progress of diethylnitrosamine-induced liver fibrosis in Sprague-Dawley rats by attenuating fibrogenesis and restricting proliferation” (ijms-870629). We have made the changes accordingly. We very much appreciate all the reviewer’s suggestions.
|
Comments and suggestions |
Changes made to the text |
|
1. There are some places that incorrectly and inaccurately write down the manuscript such as "Figure 1A" or "Fig. 4A", Line number 308 "4.7 Western blot analysis" and Line number 337 "4.7 Statistical Analysis" (Same number). Many of places need to be correct. Authors need to pay attention to check of this manuscript. ·
|
Thanks. The corrections were made accordingly. |
|
2. Please describe more detail about Figure 1A otherwise it is difficult to distinguish between the pictures. · |
Figure 1 was changed for better visualization and the text was improved (line 105-109), also supplementary figure 2 images were added for H&E and Masson with the intention to support the description of the results.
|
|
3. In Figure 4C, it is hard to say that the cleaved caspase 3 expression was different between control and DEN group because the expression of cleaved caspase 3 was too high in control group. 4. In figure 4F, Bcl-2 expression looks not different between DEN only and RvE1 samples. Based on the data (quantified with beta-actin and Bcl-2), DEN+RvE1 sample compare to RvE1 only.
|
We improved the images 4 and 5 related to western blot: added molecular weight, changes images for a better visualization. Also, it was corrected the error “relative” to the normalization to housekeeping realized. |
|
4(5). In Figure 5A, I strongly suggest that authors have to reloading samples on the same gel and need to show the same blot eventhough authors showed the GAPDH and Histone H1. Otherwise, it is hard to say that the extraction went well. |
Unfortunately, due to Covid-19 pandemic, our Laboratory and University are under lockdown and we cannot afford to perform new experiments. But as we described to Reviewer 1, we used all the internal controls to ensure no contamination in the process of extraction. |
Round 2
Reviewer 1 Report
Author done their best in lockdown time to improve the paper, maybe this limitation could be noted in the paper.
Author Response
As a team we appreciate your help.Reviewer 2 Report
No further comments.
Author Response
Thanks for your time. As a team we appreciate your helpReviewer 3 Report
The authors answered all my questions.
The manuscript is enough to publish.
Thanks.
Author Response
Thank you very much. We appreciate your help